# In situ copper faceting enables efficient CO₂/CO electrolysis

Kaili Yao[1,2,10], Jun Li [3,10] ✉, Adnan Ozden[4,10], Haibin Wang [1,10], Ning Sun[3], Pengyu Liu[3], Wen Zhong[3], Wei Zhou[5], Jieshu Zhou[1], Xi Wang[1], Hanqi Liu[3], Yongchang Liu [1,6], Songhua Chen[7], Yongfeng Hu [8], Ziyun Wang [9], David Sinton [4] ✉ & Hongyan Liang [1] ✉

The copper (Cu)-catalyzed electrochemical CO₂ reduction provides a route for the synthesis of multicarbon (C₂₊) products. However, the thermodynamically favorable Cu surface (i.e. Cu(111)) energetically favors single-carbon production, leading to low energy efficiency and low production rates for C₂₊ products. Here we introduce in situ copper faceting from electrochemical reduction to enable preferential exposure of Cu(100) facets. During the pre-catalyst evolution, a phosphate ligand slows the reduction of Cu and assists the generation and co-adsorption of CO and hydroxide ions, steering the surface reconstruction to Cu (100). The resulting Cu catalyst enables current densities of > 500 mA cm⁻² and Faradaic efficiencies of >83% towards C₂₊ products from both CO₂ reduction and CO reduction. When run at 500 mA cm⁻² for 150 hours, the catalyst maintains a 37% full-cell energy efficiency and a 95% single-pass carbon efficiency throughout.

Renewable electricity powered CO₂ reduction (CO₂R) to multicarbon (C₂₊) products is a promising approach to carbon recycling, and an attractive alternative to fossil fuel reliant pathways. Copper is the only monometallic catalyst capable of catalyzing CO₂ to C₂₊ products, and its crystal facets have been known to exert a significant influence on the CO₂R activity and selectivity[1–4]. Specifically, the thermodynamically favorable Cu(111) facet favors C₁ production, whereas Cu(100) is active for C-C coupling[2,3,5]. To date, previous works in facet-controlled synthesis of Cu(100) catalysts mostly relied on colloidal synthesis using capping agents[6–8] that modulate the relative energy of facets during synthesis. However, organic additives often play impact the catalytic performance[9] and the resulting well-defined Cu catalysts such as Cu(100) nanocubes are prone to reconstruct during electrolysis[7,10,11].

One approach to bridge the demands of catalyst fabrication and performance testing is to use the same intermediate species throughout. This improves compatibility between synthesis and catalysis, and potentially sidesteps the challenges of catalyst reconstruction during electrolysis. To realize such an approach, the electrochemical reduction of Cu-based compounds offers one avenue to enable in situ growth of active sites[12–15].

CO₂ molecules are easily activated on the catalyst surface under negative potentials, in which the reduced intermediate species, such as CO, has strong interaction with Cu[16,17] and can play a role analogous to that of capping agents to direct the in situ growth of active Cu(100) during CO₂R[13]. However, in situ reduction of traditional copper compounds is relatively fast and typically completed in less than five minutes[13,14,18,19] – an insufficient time for generating reduced

[1]School of Materials Science and Engineering, Tianjin University, Tianjin 300350, China. [2]Faculty of Chemical Engineering, Kunming University of Science and Technology, Kunming 650500, China. [3]Frontiers Science Center for Transformative Molecules, School of Chemistry and Chemical Engineering, and Zhangjiang Institute for Advanced Study, Shanghai Jiao Tong University, Shanghai 200240, China. [4]Department of Mechanical and Industrial Engineering, University of Toronto, 5 King's College Road, Toronto, ON M5S 3G8, Canada. [5]School of Science, Tianjin University, Tianjin 300350, China. [6]State Key Lab of Hydraulic Engineering Simulation and Safety, School of Materials Science and Engineering, Tianjin University, Tianjin 300354, China. [7]College of Chemistry and Material Science, Longyan University, Longyan 364012, China. [8]Sinopec Shanghai Research Institute of Petrochemical Technology, Shanghai 201208, China. [9]School of Chemical Sciences, The University of Auckland, Auckland 1010, New Zealand. [10]These authors contributed equally: Kaili Yao, Jun Li, Adnan Ozden, Haibin Wang. ✉e-mail: lijun001@sjtu.edu.cn; sinton@mie.utoronto.ca; hongyan.liang@tju.edu.cn

intermediates at high coverages to direct crystal growth. As a result, the thermodynamically favorable Cu(111) facet is preferably exposed over Cu(100) (Fig. 1a), and $C_{2+}$ production rates are low (<200 mA cm$^{-2}$) from $CO_2$R operated in a zero-gap membrane electrode assembly (MEA) electrolyzer[13,18]. Technoeconomic assessments suggest that production rates exceeding 300 mA cm$^{-2}$ are essential for commercial viability[20–22], and motivate a new approach to catalyst synthesis.

We took the view that hydroxide ions on or near the copper surface can lower the $CO_2$ activation energy barrier and facilitate CO adsorption near the OH moiety[23–25]. Introducing OH$^-$ in a Cu pre-catalyst has also been shown to promote the growth of Cu(100)[26], yet

rapid OH$^-$ loss from the Cu surface via electroreduction results in a surface with Cu(100) and Cu(111) facets in similar proportion. We posited that the buildup of stable CO and OH$^-$ (CO&OH$^-$) on Cu could promote the in situ synthesis of Cu(100)-rich catalysts compatible with efficient $CO_2$ electrolysis to multicarbon products (Fig. 1b).

Here we report an in situ copper faceting strategy to enable preferential exposure of Cu(100), accelerating electrosynthesis of $C_{2+}$ from both $CO_2$R and COR. We introduce a phosphate ligand that facilitates CO&OH$^-$ co-adsorption on Cu during electrolysis. The subsequent interactions between CO&OH$^-$ and Cu guide the surface evolution, thereby favoring Cu(100) growth. We integrate the resulting Cu(100)-rich catalyst into a MEA electrolyzer and demonstrate a $CO_2$-

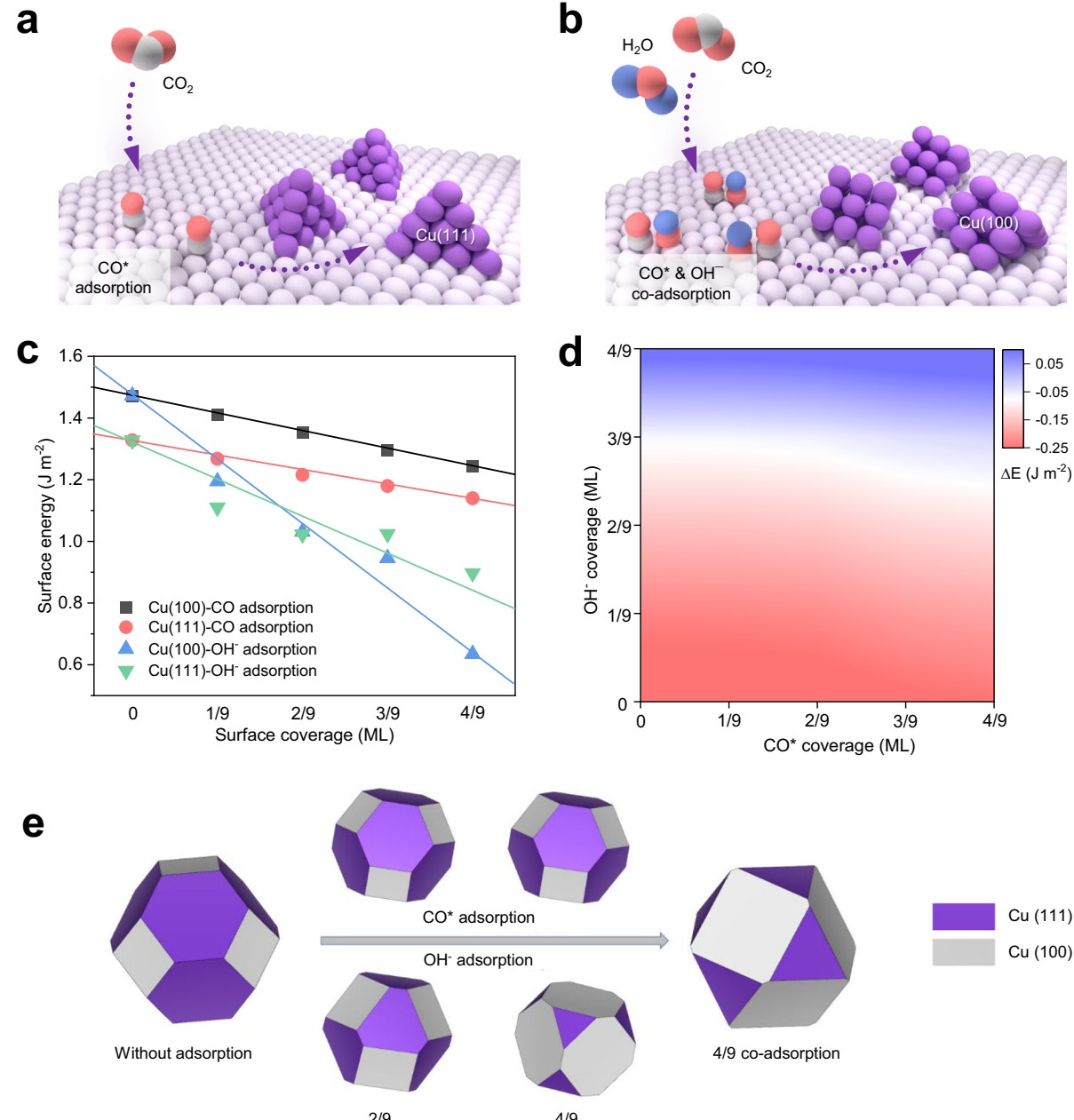

**Fig. 1 | Schematic illustration and DFT prediction of the Cu faceting process.**
**a** Schematics of the in situ Cu(111) formation from weak CO adsorption during $CO_2$R. **b** Schematics of the in situ Cu(100) formation from the co-adsorption of CO&OH$^-$ during $CO_2$R. **c** The surface energies of Cu(111) and Cu(100) at different CO* and OH$^-$ surface coverages. **d** A contour plot of the surface energy difference between Cu(100) and Cu(111) by changing the surface coverages of CO* and OH$^-$. **e**, Wulff construction clusters of copper without and with (co-)adsorption of CO* and OH$^-$.

to-$C_{2+}$ conversion system with a Faradaic efficiency (FE) of 83% at 500 mA cm$^{-2}$. The catalyst also shows excellent performance from CO electrolysis, which enables single-pass carbon efficiency (SPCE) of 95%, full-cell energy efficiency (EE) of 41%, FE of 93%, and partial current density of 465 mA cm$^{-2}$ towards $C_{2+}$ products. The Cu(100)-rich catalyst is also stable, providing sustained performance for an initial 150 operating hours.

## Results

### Density functional theory (DFT) calculations

We began by assessing the surface coverage effect of CO* and OH$^-$ on the surface energies of Cu(111) and Cu(100) using DFT (Fig. 1c, Supplementary Figs. 1–6), in which there is no adsorbate-adsorbate interaction when calculating the surface energy. For the bare Cu, the calculated surface energies of Cu(111) and Cu(100) are 1.33 J m$^{-2}$ and 1.47 J m$^{-2}$, respectively, suggesting that Cu(111) is the most stable facet in polycrystalline Cu, consistent with previous reports[13,27]. With the increase of CO* coverage, we found that the surface energies of Cu(111) and Cu(100) were both decreased, in which Cu(111) was favored over Cu(100) across a broad range of CO coverage from 0 ML to 4/9 ML. Thus despite an increase in the proportion of Cu(100) coverage, Cu(111) remains dominant in copper catalysts that are in situ synthesized using single $CO_2R$ intermediates (e.g. CO) – a finding consistent with the literature[13,18,19].

In contrast, the effect of OH$^-$ on Cu(100) formation is distinct; at an OH$^-$ coverage exceeding 2/9 ML, Cu(100) growth became more favorable over Cu(111) formation. Accordingly, we found that the adsorption energy of OH$^-$ is more negative than that of CO* (Supplementary Table 1), indicating that OH$^-$ adsorption on Cu is more conducive to regulating the surface energy of the copper catalyst. Moreover, Cu(100) has a lower OH$^-$ adsorption energy than Cu(111) due to a lower coordination number of surface atoms. Additional electrons of OH$^-$ do not affect the dipole moment and magnetic property of Cu (Supplementary Fig. 7 and Supplementary Tables 2–4). The OH$^-$ adsorption therefore may not only decrease the surface energy of the system but also preferably promote the growth of Cu(100). The projected density of states about Cu(111) and Cu(100) with different *CO and OH$^-$ coverage were calculated to provide the electronic analysis for the adsorbates' effect on surface energies. We found that the d band center of copper downshifts with increased adsorbate coverage (Supplementary Fig. 8), indicating that the *CO and OH$^-$ species could stabilize the surface.

We then generated a surface energy difference (ΔE) map for Cu(111) and Cu(100) relating to CO* and OH$^-$ coverages (Fig. 1d). We showed that with the buildup of CO&OH$^-$ co-adsorption, the growth of Cu(100) was prioritized relative to Cu(111) formation. We also carried out the Wulff construction calculation to estimate the ratio of Cu(100) and Cu(111) of a copper crystal with different coverages of CO* and OH$^-$ (Fig. 1e). At CO* and OH$^-$ coverages of 4/9 ML, we obtained a 95% increase in the Cu(100) portion relative to copper without intermediates (Supplementary Table 5).

### Synthesis and characterizations of precatalyst and derived Cu catalyst

Motivated by our DFT analysis, we sought to synthesize Cu(100)-rich catalysts with the aid of CO&OH$^-$ co-adsorption. To realize this, we turned our attention to phosphate anions, which bind strongly with Cu cations to form stable Cu complexes and can act as proton transport mediators to facilitate electrolysis[28–30]. We hypothesized that the electrochemical reduction of Cu precatalyst would be slowed by the addition of phosphate, which can promote the local proton concentration[29] and facilitate $CO_2R$ towards the co-production of CO* and OH$^-$ species; these active species adsorb easily at the positively charged Cu sites and thereby steer the Cu reconstruction to forming active Cu(100) facets (Fig. 1b)[24–26].

Seeking experimentally to coordinate phosphate anions to Cu, we prepared a phosphate-doped copper oxychloride precatalyst using a sol-gel synthesis method. We modulated the P/Cu atomic ratio up to 0.6 in the Cu precatalysts by adjusting the concentrations of phosphate and copper precursors (Supplementary Fig. 9 and Supplementary Table 6), evidenced by energy-dispersive X-ray spectroscopy (EDX) and inductively coupled plasma optical emission spectroscopy (ICP-OES). Scanning transmission spectroscopy (SEM) and transmission electron microscopy (TEM) results showed that the precatalysts with or without phosphate showed a similar particle size of ~1 μm (Supplementary Figs. 10–12).

To characterize the electronic structure of phosphate-containing Cu precatalyst, we carried out the Cu K-edge X-ray absorption spectroscopy (XAS) measurements. The X-ray absorption near edge structure (XANES) and its first derivative, Fourier-transformed extended X-ray absorption fine structure (EXAFS), together with the wavelet transform contour plot spectra suggested that the precatalyst exhibits a Cu structure akin to CuO (Fig. 2a and b, Supplementary Fig. 13). The phosphate doping in the precatalyst was further evidenced by X-ray photoelectron spectroscopy (XPS, Supplementary Fig. 14). In addition, X-ray diffraction (XRD) and Fourier transform infrared spectroscopy (FTIR) results suggested that the phosphate-doped Cu precatalyst has a crystal structure matching the orthorhombic $Cu_2(OH)_3Cl$[31], with the addition of phosphate ligands forming mono- or bidentate- complexes with the Cu cations (Fig. 2c, Supplementary Figs. 15–17).

Seeking to investigate the catalyst-evolution process, we spray-coated the precatalyst on a gas diffusion layer which was then fully reduced under $CO_2R$ conditions. The full reduction is evidenced by additional XPS analysis (Supplementary Fig. 18) and the color change of the electrode from green to metallic Cu red as seen in Raman microscope images (Fig. 2d–f). Concurrently, the precatalyst underwent a dissolution-redeposition process, resulting in a significant morphological reconstruction from particles to cavities and dendrites. XRD and dark-field TEM results showed the predominant Cu(100) over Cu(111) facets in the resulting Cu catalyst (Fig. 2c and g, Supplementary Fig. 19). By contrast, a Cu(111)-rich catalyst was obtained by reducing the phosphate-free Cu precatalyst under the same conditions (Fig. 2c, Supplementary Figs. 20 and 21). Additional electrochemical OH$^-$ adsorption measurements further indicate that Cu(100)-rich and Cu(111)-rich catalysts were derived from precatalysts with and without phosphate ligands, respectively (Supplementary Fig. 22a). These results suggest that the addition of phosphate in the Cu precatalyst favors the growth of Cu(100) during $CO_2R$ and the portion of Cu(100) of derived Cu catalyst is increased by ~95% compared to that of a previous Cu catalyst synthesized in situ[13] (Supplementary Table 7) and that of the bare Cu control catalyst (Supplementary Figs. 15b and 22b, c, and Supplementary Table 8), consistent with our DFT simulations.

### In situ spectroscopic studies of catalyst evolution

To gain molecular-level insights into the impact of phosphate ligands on Cu(100) formation, we conducted time-dependent in situ Raman experiments over the course of 30-min reduction time (Fig. 3a, Supplementary Figs. 23 and 24). For the phosphate-doped precatalyst, the intense peak at 298 cm$^{-1}$ is attributed to the vibrations of the $PO_4^{3-}$ group[32]. The phosphate peak intensity gradually decreased over the first 20 min. Concurrently, two new peaks at ~510 cm$^{-1}$ and ~2060 cm$^{-1}$ emerged at 3 min attributable to Cu-OH$_{ad}$ and CO* stretching, respectively[33,34], and these two intense peaks persisted thereafter throughout the catalyst-evolving process. The high coverage of CO* at the catalyst surface was further verified by in situ attenuated total reflectance-surface enhanced infrared absorption spectroscopy (ATR-SEIRAS, Supplementary Fig. 25). By contrast, the phosphate-free precatalyst underwent a complete reduction within 10 mins with only weak OH$^-$ and CO* adsorptions present on Cu over this period (Fig. 3a). This result supports the notion that phosphate doping slows the initial

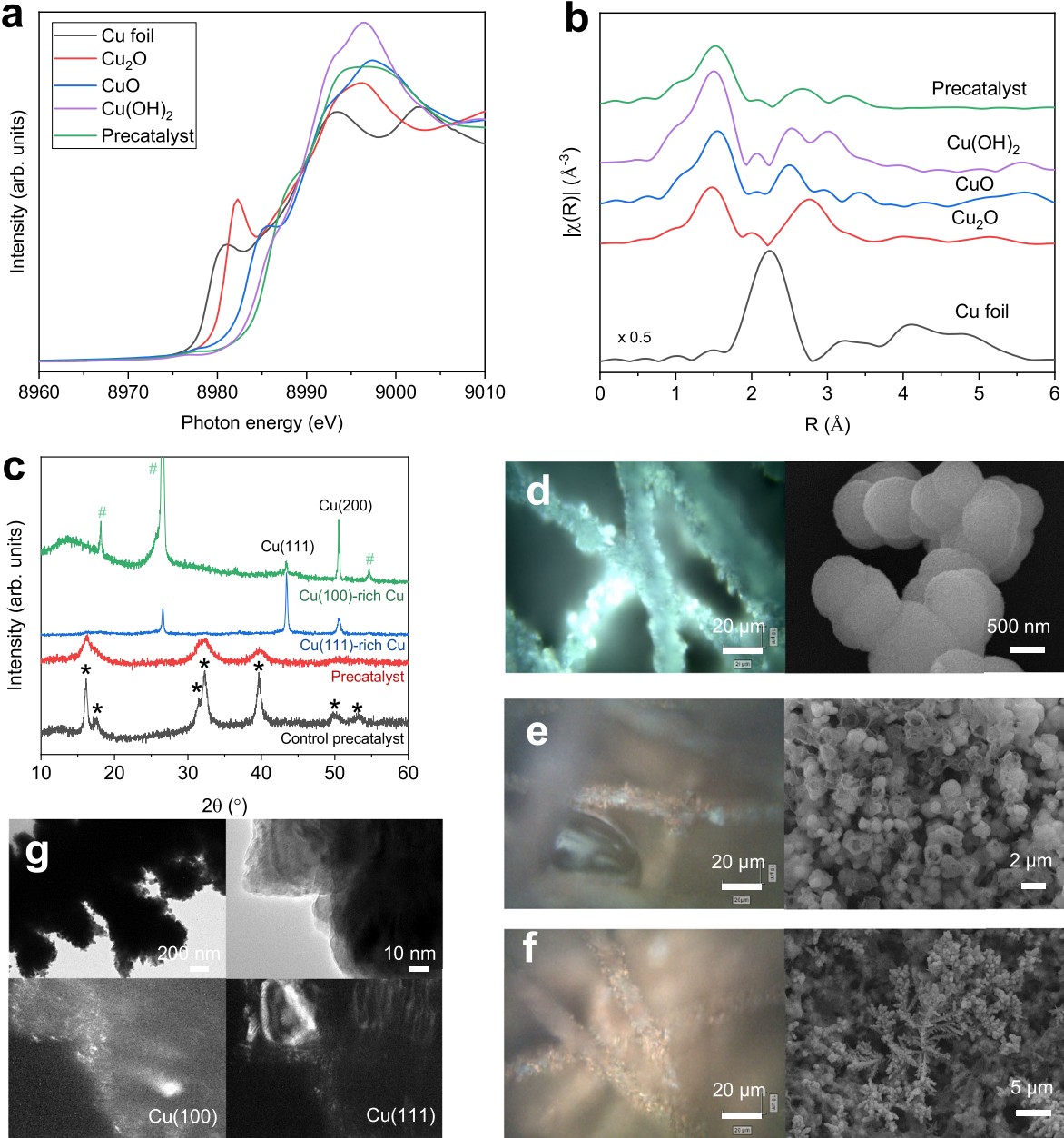

**Fig. 2 | Structural characterization of Cu precatalyst and derived Cu catalyst.** The Cu K-edge XANES (**a**) and Fourier-transformed EXAFS (**b**) spectra of Cu precatalyst and standards (Cu foil, $Cu_2O$, CuO, and $Cu(OH)_2$). **c** XRD spectra of precatalysts with and without phosphate and the relevant derived Cu catalysts after reduction; * and # represent diffraction peaks of $Cu_2(OH)_3Cl$ and carbon paper

substrate, respectively. Dark-field microscope images (left) and related SEM images (right) of Cu precatalyst after electrolysis at different reduction times, i.e. 0 min (**d**), 15 min (**e**), and 30 min (**f**). **g** Bright-field (top) and dark-field (bottom) TEM images of the Cu(100)-rich catalyst.

reconstruction of Cu and enables the enhanced co-adsorption of CO&OH⁻ assisted by phosphate doping.

To examine the catalyst formation process at an atomic level, we performed the time-dependent in situ Cu K-edge XAS experiments to monitor the dynamic changes of the Cu oxidation state and local coordination environment. During $CO_2R$, the Cu precatalyst was fully reduced in 30 mins, in which the whiteline intensity gradually decreased whereas the pre-edge feature at ~8981 eV was progressively enhanced (Fig. 3b). Two isosbestic points at 8995 eV and 9005 eV in the XANES region were observed, indicating a two-phase transformation from the Cu precatalyst to form a metallic Cu structure[35,36]. This finding is supported by EXAFS results (Fig. 3c), in which the intensity of Cu−O bond slowly decreased with the emergence of metallic Cu-Cu bond.

We then fit the XANES spectra using a linear combination method[14,19,37] (Fig. 3d and e, Supplementary Table 9). We found that the catalyst evolution went through an exponential-like process: the Cu species underwent a rapid reduction at the first 15 mins corresponding to precatalyst dissolution (Fig. 2e, Supplementary Fig. 26); then at the catalyst redeposition stage, the Cu reduction assisted by co-adsorption of CO&OH⁻ was relatively slow (Figs. 2f and 3a, Supplementary Fig. 26). This was further confirmed by the EXAFS fitting results[37], in which the formation of metallic Cu-Cu bond also followed an exponential-like trend (Fig. 3f, Supplementary Table 10). The covalent Cu-O coordination exhibited a steep reduction (Fig. 3g), suggesting a relatively steady O-leaching from the precatalyst matrix during chronoamperometry electrolysis. We also performed in situ Cu

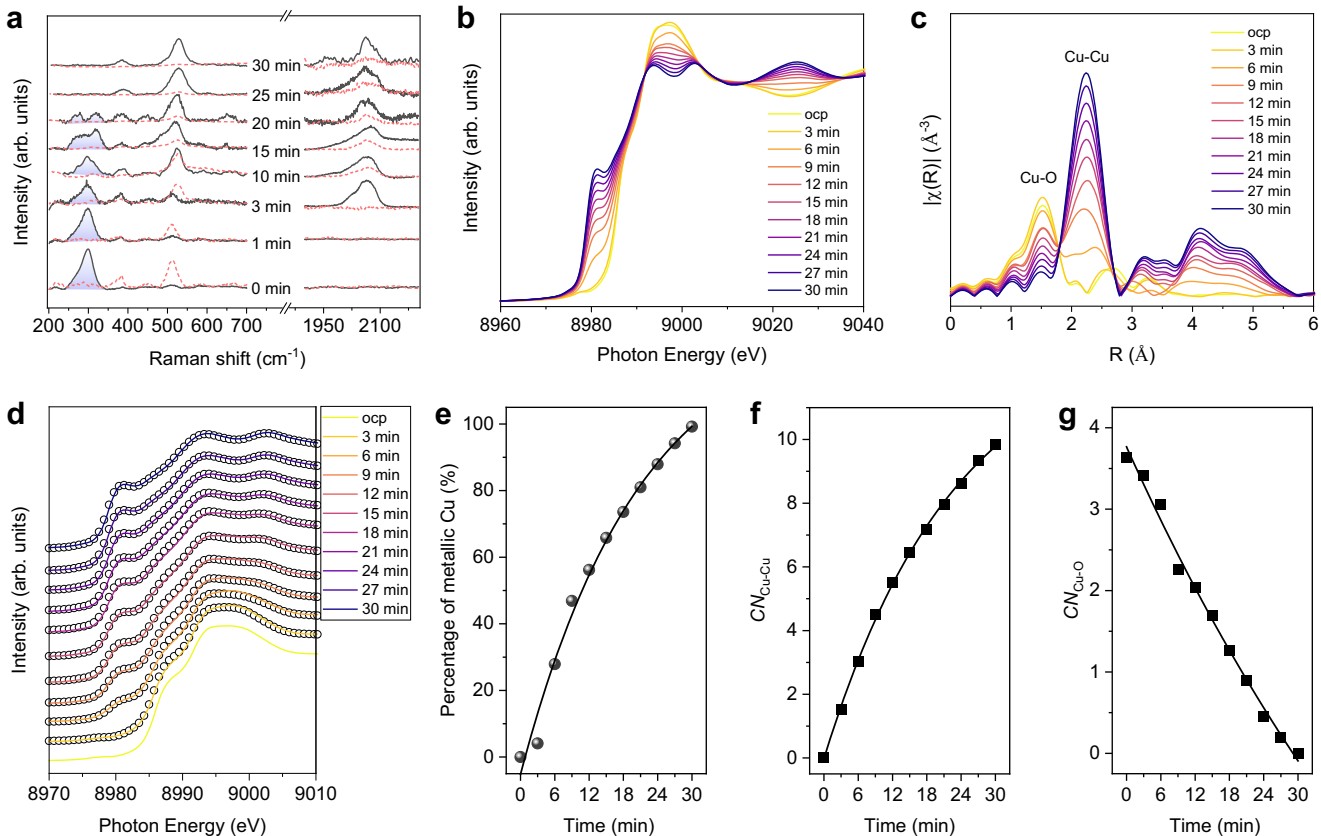

**Fig. 3 | In situ spectroscopic interrogation. a–c** In situ Raman spectra of Cu precatalyst evolution with (solid) and without (dot) phosphate collected in $CO_2$-flowed 0.1 M $KHCO_3$ electrolyte at −1.1 V *vs* RHE; the shade areas represent the vibrations of the $PO_4^{3-}$ group[32]. Time-dependent Cu K-edge XANES (**b**) and Fourier-transformed EXAFS (**c**) of Cu precatalyst with phosphate addition, the test was performed in $CO_2$-flowed 0.1 M $KHCO_3$ electrolyte at −1.1 V vs. RHE over the course of 30-min reduction time; ocp stands for open-circuit potential. **d** The corresponding XANES fitting spectra in **b**, solid lines represent the experiment data and circles represent the linear combination fit spectra. **e** The percentage of metallic Cu at different reduction times. Values are extracted from linear combination fit using the XANES spectra of Cu precatalyst at ocp and metallic Cu as standards with the weighting factors as fit parameters. Coordination numbers (CN) of Cu-Cu (f) and Cu-O (**g**) at different reduction times extracted from the Cu K-edge EXAFS fittings (see Supplementary Table 10).

K-edge XAS analysis of the control Cu precatalyst without phosphate (Supplementary Fig. 27); it shows a rapid Cu reduction compared to the Cu precatalyst with phosphate. These results also indicate that slow Cu reduction during catalyst redeposition process arises from strong noncovalent $OH^-$ adsorption on Cu[25] – an effect triggered by phosphate doping.

These findings, taken together, illustrate that the addition of phosphate enables noncovalent interactions of Cu with $OH^-$ and CO* to prioritize the growth of Cu(100)-rich catalyst, promising more efficient $C_{2+}$ production.

### Electrochemical performance of $CO_2$ and CO reductions

We evaluated the $CO_2R$ performance in a catholyte-free MEA electrolyzer with 0.1 M $KHCO_3$ flowing at the anode. Cu(100)-rich catalyst shows a peak $FE_{C2+}$ of 83% at 500 mA $cm^{-2}$ with a full-cell voltage of −3.8 V, translating to an $EE_{C2+}$ of 25% and a $C_{2+}$ current density of 415 mA $cm^{-2}$ (Fig. 4a). This performance (EE × current) is double that of reported results in neutral MEA-$CO_2R$ systems (Supplementary Fig. 28 and Supplementary Tables 7 and 11), indicating the role of Cu(100)-rich sites in accelerating C-C coupling. The effect of electrochemical surface area (ECSA) on $CO_2R$ reactivity is not significant: Cu(100)-rich catalyst showed an ECSA value akin to Cu(111)-rich catalyst derived from a precatalyst without phosphate (Supplementary Fig. 29 and Supplementary Table 12). However, the Cu(100)-rich catalyst exhibited a 1.7× higher normalized $C_{2+}$ current density than the Cu(111)-rich catalyst, suggesting the intrinsic nature of activity improvement on Cu due to increased Cu(100). The Cu(100)-rich catalyst is stable, achieving a $FE_{C2+}$ of 75% and an $EE_{C2+}$ of 24% for initial 60 hours at 300 mA $cm^{-2}$ (Fig. 4b).

Increasing CO and $OH^-$ concentrations in the catalyst layer would enhance the adsorptions of $OH^-$ and *CO[38] to facilitate $C_{2+}$ production on the Cu(100)-rich catalyst. Accordingly, we evaluated the performance of the Cu(100)-rich catalyst in the same MEA system by feeding CO gas at the cathode and flowing 1 M KOH at the anode: it shows a peak $FE_{C2+}$ of 93% and a $C_{2+}$ current density of 465 mA $cm^{-2}$ at −2.3 V, leading to an $EE_{C2+}$ of 41% (Fig. 4c). In situ Raman analysis (Supplementary Fig. 30) evidenced the co-adsorption of *CO&$OH^-$ at the catalyst with vastly increased $OH^-$ adsorption in the CO reduction system compared to the $OH^-$ adsorption during $CO_2$ electrolysis. This finding is consistent with our DFT results that additional $OH^-$ adsorption helps stabilize Cu(100) to accelerate C-C coupling.

To advance $SPCE_{C2+}$ in CO electrolysis, we decreased the areal flow rate of CO to 2 sccm $cm^{-2}$. At a current density of 500 mA $cm^{-2}$, we achieved an SPCE of 95% for $C_{2+}$ products (Fig. 4d). The Cu(100)-rich catalyst is robust, delivering an average $SPCE_{C2+}$ of 95%, a $FE_{C2+}$ of 86%, an $EE_{C2+}$ of 37%, and a $C_{2+}$ current density of 432 mA $cm^{-2}$ for 150 hours (Fig. 4e). This outperforms previously reported MEA-CO electrolysis systems with respect to EE × current (Supplementary Fig. 31 and Supplementary Table 13).

In summary, this work presents a strategy to stabilize intermediates that facilitate Cu(100) growth during in situ catalyst materials synthesis. Using this strategy, we achieved efficient $C_{2+}$ productions

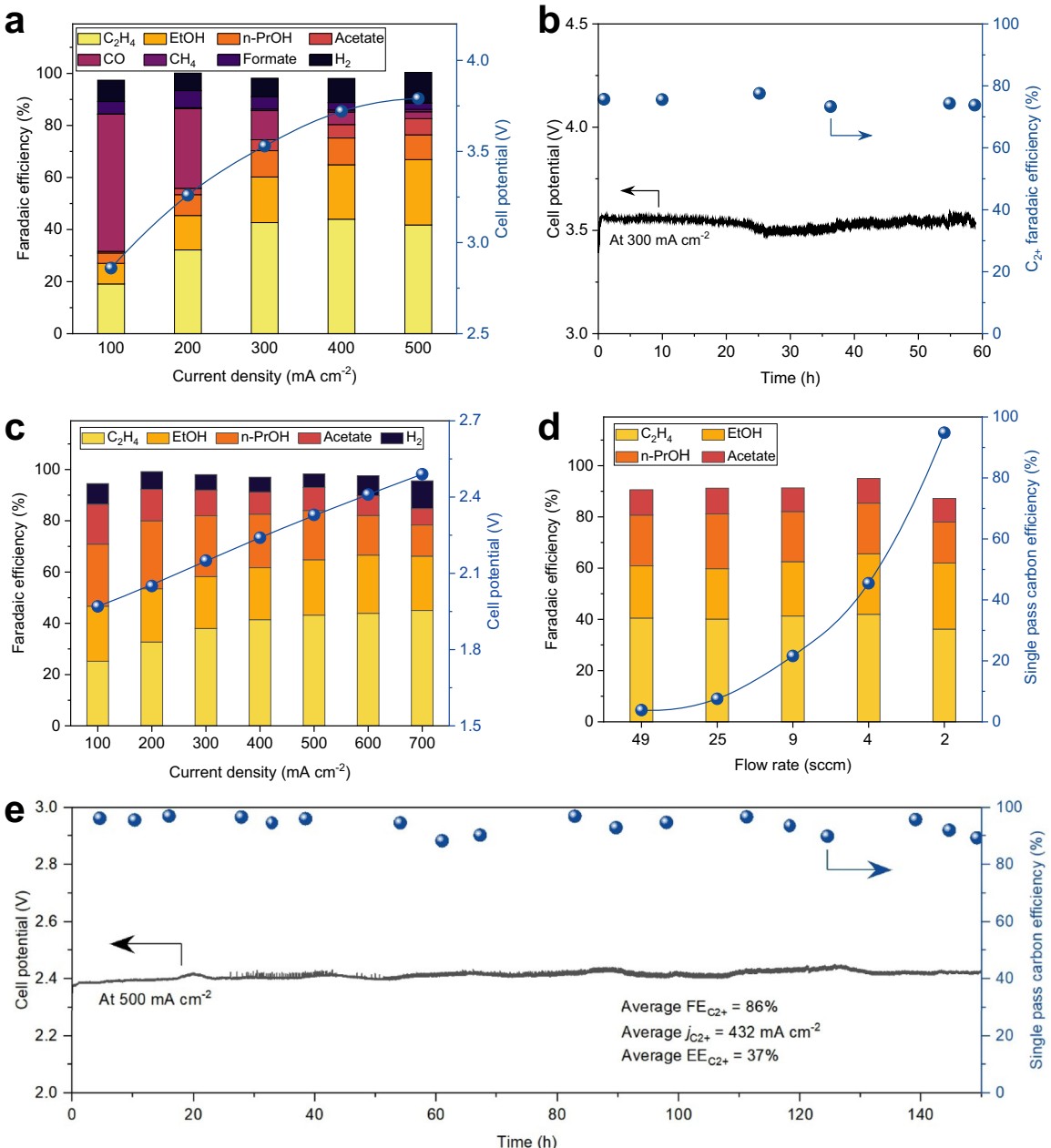

**Fig. 4 | Electrochemical CO₂/CO reduction performance of the Cu(100)-rich catalyst in a MEA electrolyzer. a** FEs for all products at various current densities and related cell potentials from $CO_2$ reduction. **b** Extended $CO_2$ electrolysis using the Cu(100)-rich catalyst at 300 mA cm⁻². Operating conditions: 0.1 M $KHCO_3$ anolyte (pH -8.4) at a flow rate of 20 mL min⁻¹ and an average $CO_2$ inlet flow rate of -75 sccm cm⁻² at atmospheric temperature and pressure conditions, in which the catalyst loading is -1 mg cm⁻². **c** FEs for all products at various current densities and related cell potentials from CO reduction. **d** FEs as well as SPCEs toward $C_{2+}$ products on the Cu(100)-rich catalyst at 500 mA cm⁻² with different CO flow rates. **e** Extended CO electrolysis using the Cu(100)-rich Cu catalyst at 500 mA cm⁻². The Cu(100)-rich catalyst delivers an average $FE_{C2+}$ of 86%, $C_{2+}$ current densities of 432 mA cm⁻², $EE_{C2+}$ of 37%, and $SPCE_{C2+}$ of 94% throughout. Operating conditions: 1 M KOH anolyte (pH -14) at a flow rate of 20 mL min⁻¹ and an average CO inlet flow rate of -2 sccm cm⁻² at atmospheric temperature and pressure conditions, in which the catalyst loading is -1 mg cm⁻².

from CO₂/CO electrolysis with a high energy efficiency of 40% and a near-unity carbon efficiency. This performance is in the vicinity of industrial standards[20,39] and brings forward this technology in the direction of economic viability.

## Methods
### Precatalyst preparation
The precatalysts were fabricated by the following sol-gel method: first, copper (II) chloride dihydrate ($CuCl_2 \cdot 2H_2O$, 0.4104 g) was dissolved in 2 mL methanol, sodium hypophosphite ($NaH_2PO_2$) in 2 mL methanol

was also prepared in a separate vial, two solutions mentioned above were cooled in the freezer for 2 h. Next, $NaH_2PO_2$ was then added dropwise to $CuCl_2 \cdot 2H_2O$. Then, 1 mL epoxypropane and a mixture of methanol and water (2 mL MeOH and 0.23 mL $H_2O$) were slowly added under constant stirring resulting in a blue solution. The solution was then allowed to age at room temperature for more than three days to promote network formation and gelation. After aging, the gels were repeatedly washed and centrifuged with acetone three times and then dried under a vacuum for 24 hours. The precursors with different phosphorus-doped concentrations ($NaH_2PO_2$, 0.0306, 0.0918, and

0.1530 g) were synthesized by changing the amount of $NaH_2PO_2$ and were denoted as CuP0.2, CuP0.4, and CuP0.6, respectively. As a control, the precursor without phosphorus was also synthesized, labeled CuP0.

## Electrode preparation

To prepare the deposition ink, 20 mg of the precursor was dispersed in a mixture of 0.95 mL 2-propanol, and 50 µL 5 wt% Nafion solution (Sigma-Aldrich) and then sonicated for at least 1 hour. The ink was air-brushed onto 2 × 4 cm carbon paper (Toray TGP-H-060 with an MPL layer) with the loading of ~1 mg/cm² and dried to form the copper sol-gel working electrode. The loading was determined by measuring the weight of the carbon paper before and after deposition.

## CO₂R in flow-cell

The $CO_2R$ activity of the catalysts was investigated by performing electrolysis in a two-compartment flow-cell configuration using a 1.0 M KOH electrolyte. The three-electrode set-up was connected to a potentiostat (Autolab 204). Flow-cell configuration consists of catalysts spray-coated gas diffusion layer as the cathode, an anion exchanged membrane (AEM, Fumasep FAA-3-PK-130) as the membrane, and NiFe/Ni foam as the anode, Ag/AgCl was used as the reference electrode. $CO_2$ gas flowed behind the gas diffusion layer at a rate of 50 mL/min. The performance of the cathode was evaluated by performing constant-current electrolysis.

Gas products were routed into a gas chromatograph with a thermal conductivity detector and a flame ionization detector (GC-2060A). Nitrogen (99.999%) was used as the carrier gas. ¹H NMR spectrum (Bruker, AVANCE IIITM HD 400 MHz) was used to determine the liquid products in 10% $D_2O$ using water suppression mode, for which dimethyl sulfoxide was added as an internal standard. All the electrode potentials were converted to values regarding the RHE using the following equation:

$$E_{RHE} = E_{Ag/AgCl} + 0.197V \times 0.0591 \times pH + 85\% \, iR \qquad (1)$$

The ohmic loss between the working and reference electrodes was measured using the electrochemical impedance spectroscopy technique (with frequency ranges from 100 kHz to 0.1 Hz and amplitude of 10 mV.) and 85% $iR$ compensation ($i$ is the current density, $R$ is the uncompensated resistance) was applied to correct the potentials manually.

## CO₂R/COR in MEA electrolysers

The $CO_2R$ and COR experiments were performed in a membrane electrode assembly (MEA) using neutral (0.1 M $KHCO_3$) and alkaline (1 M KOH) media electrolytes, respectively. The performance tests were conducted using an electrochemical test station. The test station was equipped with the components, including a commercial MEA electrolyzer, a current booster (Metrohm Autolab, 10A), a mass flow controller (Sierra, SmartTrak 100), a humidifier, an electrolyte container, and a peristaltic pump with silicon tubing. The MEA setup consisted of titanium anode and stainless-steel cathode flow-field plates with geometric flow-field areas of 1 cm². The MEA was composed of three components: a cathode electrode (as described earlier), an anode electrode (IrO$_x$ supported on a titanium (Ti) felt (Fuel Cell Store)), and an anion exchange membrane (AEM, Sustainion® X37-50). The cathode and anode electrodes were separated from each other using an AEM. The AEM was rinsed with DI water for 15 min before performance testing. The MEA was assembled by a compression torque applied to the each of associated bolts. The cathode flow field was used for the supply of humidified $CO_2$/CO over the backside of the cathode gas diffusion electrode, whereas the anode flow field was used for the supply of anolyte through the anode electrode. The IrOx-Ti electrode was prepared by following a

procedure involving several steps: (1) immersing the Ti felts into a homogenous ink of 2-propanol, iridium (IV) chloride dehydrate (Premion®, 99.99%, metal basis, Ir 73%, Alfa Aesar), and hydrochloric acid (HCl), (2) drying the resulting electrodes at 100 °C for 10 min and sintering the dried electrodes at 500 °C for 10 min, and (3) repeating the first two steps until an IrO$_x$ mass loading of 1 mg/cm². After the MEA assembly, the anolyte (0.1 M $KHCO_3$ for $CO_2R$ and 1 M KOH for COR) was fed into the anode flow field, and the humidified gaseous reactant ($CO_2$ for $CO_2R$ and CO for COR) was fed into the cathode flow field. A constant current density of 100 mA cm⁻² was applied to initiate the $CO_2R$ or COR, and the current density was gradually increased with 100 mA cm⁻² increments. The current increments were made after stabilization of the cell potential, which typically required 30-40 minutes. At each current density, gas products of $CO_2R$ or COR were analyzed using gas chromatography (GC). The liquid products of $CO_2R$ or COR were collected from the cathodic and anodic downstream simultaneously.

## CO₂R/COR product analysis

The Faradaic efficiency of gas products was calculated as follows.

$$Faradaic\ efficiency(\%) = N \times F \times \nu \times r / (i \times V_m) \qquad (2)$$

where $N$ represents the number of electrons transferred, $F$ represents the Faradaic constant, $\nu$ represents the gas flow rate at the cathodic outlet, $r$ represents the concentration of product(s) in ppm, $i$ represents the total current, and $V_m$ represents the unit molar volume of product(s). The gas flow rate at the cathode outlet was measured via a bubble flow meter.

The liquid products of $CO_2R$ or COR were analyzed by using ¹H NMR spectroscopy (600 MHz Agilent DD2 NMR Spectrometer) with suppressing water peak. Dimethyl sulfoxide (DMSO) was used as the reference standard, whereas deuterium oxide ($D_2O$) was used as the lock solvent. The NMR spectra collected at each current density were used to calculate the Faradaic efficiency toward liquid products of $CO_2R$ or COR as follows.

$$Faradaic\ efficiency(\%) = N \times F \times n_{product} / Q \qquad (3)$$

where $N$ represents the number of electrons transferred, $F$ represents the Faradaic constant, $n_{product}$ represents the total mole of product(s), and $Q = i \times t$ represents the total charge passing during the liquid product collection.

## Energy efficiency (EE) calculation

The full-cell energy efficiencies toward $CO_2R$ or COR products were determined as follows.

$$EE_{product} = \frac{E^o_{cell}}{E_{cell}} \times FE_{product} \times 100\% \qquad (4)$$

$$E^o_{cell} = \frac{\Delta G^o}{-zF} \qquad (5)$$

where $E^o_{cell}$ represents the thermodynamic cell potential for $CO_2R$ or COR products, $\Delta G^o$ represents the change in Gibbs free energy, and $E_{cell}$ represents the applied cell voltage ($iR$-free).

## Single-pass carbon efficiency (SPCE) calculation

SPCE toward gas, liquid, or a group of gas and liquid products of COR (at 25°C and 1 atm) was calculated as follows.

$$SPCE = \frac{(j \times 60\ sec)/(N \times F)}{(flow\ rate(L/min) \times 1(min))/(24.05(L/mol))} \qquad (6)$$

where $j$ represents the partial current density toward a COR product, $N$ represents the number of electrons transferred to produce 1 mole of product.

## Material characterization

The phase of catalyst powder was verified by XRD with a Bruker D8 Advanced diffractometer using Cu Kα radiation ($\lambda = 1.5406$ Å). The surface morphology and composition of the catalysts were characterized by scanning electron microscopy (SEM, JEOL JSM-7800F) instruments with energy-dispersive X-ray spectroscopy (EDX). Transmission electron microscope (TEM), high-resolution TEM (HRTEM), and TEM-mapping images were performed on a JEOL JEM-2100F operated at 200 kV. To obtain the dark-field TEM images for the Cu(100)-rich catalyst, we first carried out selected area electron diffraction, then trapped the diffraction spots on the specific crystal facets using the aperture slot, and finally took the dark field photographs. X-ray photoelectron spectroscopy (XPS) for surface element investigation before and after $CO_2R$ was carried out by using a Thermo Scientific K-Alpha+ source XPS system. Inductively coupled plasma optical emission spectroscopy (ICP-OES, Thermo iCAP7000) was carried out to determine the contents doped into copper in precursor, 5 mg of the sample was completely dissolved into 1 L deionized water with 5 mM $HNO_3$ and sonicated for 30 min for the ICP-OES test.

## In situ X-ray absorption spectroscopy (XAS) measurements

We performed the in situ Cu K-edge XAS measurements at the Beamline BL11B of the Shanghai Synchrotron Radiation Facility and the SuperXAS beamline of the Swiss Light Source, Villigen, Switzerland. The in situ XAS setup is akin to our previous report[19]. Fluorescence X-ray signals were recorded. During measurements, XAS spectra were recorded at a time resolution of 1 s, which were then averaged for X-ray absorption near-edge spectra (XANES) and extended X-ray absorption fine structure (EXAFS) analyses. Aqueous 0.1 M $KHCO_3$ electrolytes were flowing at both the anode and cathode compartments, which were separated by an anion-exchange membrane (AEM, Fumasep FAA-3-PK-130). An Ag/AgCl (saturated KCl) and a sputtered Pt on carbon paper were used as reference and counter electrodes, respectively. A Cu precatalyst loaded on carbon paper was the working electrode. An electrochemical workstation (Gamry Interface 1000) was used to power the test.

## XAS analysis

XAS data were processed with Demeter (v.0.9.26). We performed The XAS spectral normalization was performed using Athena software in Demeter, in which a cubic spline function was used to fit the background above the absorption edge[40]. We then applied Fourier transformation of the EXAFS spectra from an energy (E) space to a radial distance (R) space. We fit EXAFS spectra using Artemis software in Demeter with the FEFF6 program. Data range $k = 3\text{-}11$ Å$^{-1}$, amplitude reduction factor $S_0^2 = 0.8$. The coordination numbers (N) were fixed to the expected values listed in cif files of $Cu_2(OH)_3Cl$ and Cu foil, bond distances (R) and the Debye Waller factor ($\sigma^2$) for each cell were determined. Then the Debye Waller values were fixed to calculate N.

## In situ Raman tests

In-situ Raman experiments were performed using a Renishaw inVia Raman microscope in a homemade $CO_2$-saturated 0.1 M $KHCO_3$ aqueous solution as the electrolyte. The signals were collected using a water-immersion objective at the excitation laser source of 633 nm.

## In situ Attenuated total reflection-surface-enhanced infrared absorption spectroscopy (ATR-SEIRAS) tests

ATR-SEIRAS were measured using a Thermo Scientific Nicolet iS50 FTIR Spectrometer with a Pike VeeMAX III attachment. 10 mg of the precursor was dispersed in a mixture of 0.95 mL 2-propanol, and 50 μL

5 wt% Nafion solution (Sigma-Aldrich) and sonicated for 30 min, then drop coated on Au film. Spectra were recorded at a resolution of 4 cm$^{-1}$ in a $CO_2^-$ saturated 0.1 M $KHCO_3$-$D_2O$ electrolyte.

## Electrochemical OH$^-$ adsorption and ECSA evaluation

Electrochemical OH$^-$ adsorption was performed in an $N_2$-saturated 1.0 M KOH electrolyte with a linear sweep voltammetry method at a sweep rate of 100 mV/s for copper. The potential ranged from −0.2 to 0.6 V $vs$ RHE. The electrochemical active surface area (ECSA) was measured by using double-layer capacitance and determined based on the equation ECSA = $R_f$S, where $R_f$ was the roughness factor and S was the geometric area of the electrode (1 cm$^{-2}$). $R_f = C_{dl}/0.034$, where $C_{dl}$ was the double-layer capacitance of catalysts and 0.029 mF/cm$^2$ was the double-layer capacitance of a smooth Cu foil[12]. All the catalysts were scanned at a non-Faradaic region of −0.06 to −0.02 V $vs$ RHE, in $N_2$-saturated 1.0 M KOH for ten cycles at sweep rates of 40, 60, 80, and 100 mV s$^{-1}$.

## DFT calculations

DFT calculations were performed using the Vienna ab initio Software (VASP)[41,42], Vander Waals interactions were accounted for by using the DFT-D3 method[43]. The generalized gradient approximation (GGA) with Perdew, Burke, and Ernzerh (PBE) functional is adopted to describe the electron exchange-correlation interaction[44,45]. Electron wave functions are expanded in plane waves with a kinetic energy cutoff of 400 eV. The method of Methfessel-Paxton (MP) with a smearing width of 0.20 eV was employed to transition metal surfaces and interfaces[46]. The convergence criterion for the electronic self-consistent iteration and force was set to 10$^{-5}$ eV and 0.05 eV Å$^{-1}$, respectively. A vacuum layer of 15 Å was introduced to avoid the interactions between periodic images. (3 × 3 × 1) k-point grid was used as these models.

To evaluate the stability of one surface, the surface energy was used as defined below[47]:

$$E_{surface} = \frac{E_{total} - nE_{ref} - E_{ads}}{2A} \tag{7}$$

Where $E_{total}$ is the total energy of this surface from DFT calculations; $E_{ref}$ is the reference energy of unit composition from bulk calculations; $E_{ads}$ is the sum of the adsorption energies of the intermediates/molecular at given coverages; A is the surface area; and $n$ is the number of unit composition in this surface, the less stable this surface is.

For each coverage of the model, the most stable configuration was selected manually. In this work, the adsorbed molecule is CO* and OH$^-$. To simulate the charge of OH$^-$, additional valence electrons were added to the model, which is equal to the number of OH$^-$. Surface energies with adsorption of two intermediates states were calculated by assuming that there were no adsorbate-adsorbate interactions[48]. The energy of CO* and OH$^-$ is obtained by the equation below:

$$E_{molecular} = E_{scf} + \Delta ZPE - T\Delta S + \int_0^{298.15K} C_p dT \tag{8}$$

Where the $E_{SCF}$ is the electronic SCF energies from DFT, $\Delta ZPE$ is the zero-point energy and $T\Delta S$ is the entropy correct. The entropy and heat capacity corrections are evaluated. For the gas molecules, rotational and translational entropies were included in addition to vibrational entropies.

## Data availability

The datasets analyzed and generated during the current study are included in the paper and its Supplementary Information, and can be obtained from the corresponding authors upon request.

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

## Acknowledgements

This work was financially supported by the Ministry of Science and Technology of China (2023YFA1507903 to H.L., 2022YFA1505100 and 2023YFA1507500 to J.L.), National Natural Science Foundation of China (51771132 and 52204320 to H.L., BE3250011 to J.L.), the Fundamental Research Funds for the Central Universities (23X010301599 to J.L.), the Shanghai Pilot Program for Basic Research - Shanghai Jiao Tong University (21TQ1400227 to J.L.), the Collaborative Innovation Platform Project of Fu-Xia-Quan National Independent Innovation Demonstration Zone (2022-P-021 to S.C.), the Ontario Research Foundation: Research Excellence Program (to D.S.), the Canada Research Chairs Program (to D.S.), the Natural Sciences and Engineering Research Council (NSERC) of Canada (to D.S.), and TOTAL SE (to D.S.). The computational study is supported by the Marsden Fund Council from Government funding (21-UOA-237 to Z.W.) and Catalyst: Seeding General Grant (22-UOA-031-CGS to Z.W.), managed by Royal Society Te Apārangi. Z.W. acknowledges the use of New Zealand eScience Infrastructure (NeSI) high-performance computing facilities, consulting support and/or training services as part of this research. H.L. acknowledges the PERIC Hydrogen Technologies Co., Ltd and Metrohm China Ltd. We acknowledge the Paul Scherrer Institut, Villigen, Switzerland, for the provision of synchrotron radiation beamtime at the beamline SuperXAS of the SLS and would like to thank M. Nachtegaal for the assistance. We also acknowledge the Shanghai Synchrotron Radiation Facility, China, for the provision of synchrotron radiation beamtime at the beamline BL11B and would like to thank J. Li and X. Zhong for the assistance.

## Author contributions

H.L., D.S. and J.L. supervised the project. K.Y. carried out the electrochemical experiments. J.L., P.L., and W.Z. performed XAS measurements and analysis with the assistance of Y.H.. K.Y. conducted Raman testing. A.O. assisted with the electrochemical measurements in the MEA electrolyzer. H.W. performed DFT calculations with the supervision of Z.W. and W.Z.. N.S. contributed to figure drafting. J.Z., X.W. and H.Liu contributed to electrochemical data collection and analysis. J.L. and K.Y. co-wrote the manuscript. D.S., H.L., Y.L. and S.C. contributed to manuscript editing. All authors discussed the results and assisted during the manuscript preparation.

## Competing interests

The authors declare no competing interests.
