## [Peer Review File · Nature Communications]

REVIEWER COMMENTS

Reviewer #1 (Remarks to the Author):

In this study, the authors investigate by DFT calculations the effect of adsorbates (CO & OH-) on the surface energies of Cu(111) and Cu(100). According to the calculation results, the author proposed OH- adsorption may not only decrease the surface energy of the system but also preferably promote the growth of Cu(100). Meanwhile, they also use Wulff construction calculation to estimate the ratio of Cu(100) and Cu(111) of a copper crystal with different coverages of CO and OH-. The results of these DFT analyses contribute to our understanding of the CO₂ reduction reaction on high-valence copper-derived catalyst systems. This study is probably publishable after the authors address the questions listed below:

1. Since the author assumed that there is no adsorbate-adsorbate interaction when calculating the surface energy, but the author did not provide the information (slab size, atomic layers) of the slab models in the computational details; that is, can the slab size of Cu(100) and Cu(111) used by the author in the calculation support the above hypothesis? Therefore, I suggest the author provide enough model details and structure diagrams for confirmation.
2. The authors should provide the adsorption structure diagrams corresponding to Table S1 and Table S2.
3. Although Tables S1 and S2 give the surface energy data for different adsorbates (CO and OH-) coverages on Cu(100) and Cu(111), if the authors can provide some electronic analysis to discuss the trends of surface energies on these surfaces would be beneficial.
4. Since the authors have added additional valence electrons to the simulation model in the DFT calculations, does this affect the dipole moment and magnetic property? If so, did the authors consider corresponding dipole moment corrections and magnetism in calculations?
5. Although the authors found by the results of DFT calculations that when CO and OH- coverage reaches 4/9 ML, the Cu(100) part will increase by 95% relative to the copper without intermediates, can the experimental conditions correspond to the results of the DFT simulation? If not, is such a coverage assumption reasonable? The authors should correlate the DFT simulation conditions with the experimental results.

Reviewer #2 (Remarks to the Author):

This paper reported the copper faceting strategy for CO₂/CO electrolysis. The fundamental study and experiments are well performed. However, the novelty of this MS does not seem sufficient for prestigious journal like NC, considering numerous publications on tuning Copper facets, including a few from the same group, such as Nat Catal 3, 98–106 (2020). In addition, the experiments and procedures developed in this study do not add any new knowledge to the existing in the field.

The authors claim high stability but Cu has been reported to go under morphology changes under cathodic conditions in the literature (e.g., Angew.Chem.Int. Ed. 2018, 57,6192 –6197), therefore the authors need to provide reasons for their stability. the highly stable cell voltage at a current density as high as 300 or 500 mA/cm² needs to be elaborated since the data reported in the literature usually comes with a level of fluctuating (which can be reasonable if current is quite high).

Reviewer #3 (Remarks to the Author):

The manuscript “In situ copper faceting enables efficient CO₂/CO electrolysis” by Yao et al reports on the enhanced CO₂RR to C₂+ products, observed on copper catalysts in the presence of phosphate ligands. Authors attribute it to the enhanced formation of Cu(100) facets during the CO₂RR in the presence of OH⁻ adsorbates, which itself seems to be facilitated by the presence of phosphate ligands. Authors also claim that their pre-catalyst features an unusually high valence state for Cu, and the presence of phosphate ligand helps to retain the high valence state of copper during the electrolysis.

While these claims are intriguing, unfortunately, I do not see strong evidences for most of them. First, the central claim of the manuscript is that the presence of phosphate stimulates formation of Cu(100) facets over Cu(111). However, nowhere in the manuscript I can see a quantitative analysis that this is indeed the case. Authors only claim that “XRD and dark-field TEM results showed the predominant Cu(100) over Cu(111) facets in the resulting HVCD Cu catalyst (Figs. 2d and 2h, and Supplementary Fig. 11). By contrast, a Cu(111)-rich catalyst was obtained by reducing the phosphate-free Cu precatalyst under the same conditions (Fig. 2d and Supplementary Figs. 12-13).” From the provided images I could agree that the morphology of the catalyst after the experiment are different, if the CO₂RR is carried with or without phosphate. However, I do not see any convincing, quantitative analysis that would evidence that indeed one of these catalysts have more Cu(100) facets than the other. Similarly, authors also state that “Additional electrochemical OH⁻ adsorption measurements further indicate that Cu(100)-rich and Cu(111)-rich catalysts were derived from precatalysts with and without phosphate ligands, respectively (Supplementary Fig. 14).” However, it is not clear to me how such quantitative statement can be easily extracted from Figure S14.

The claim that the pre-catalyst features Cu sites in oxidation state higher than 2+ is also not supported by the data, in my opinion. Note that this authors' conclusion is based on XAS data, where authors compare the Cu K-edge XAS spectra for their catalyst with those of metallic Cu, Cu₂O and CuO. However, the shape of XAS features depends not only on the Cu oxidation state, but also on the local structure. If authors would compare the XAS data for their catalyst with, e.g., the reference spectrum for Cu(OH)₂, where Cu is obviously in Cu²⁺ state, they would see that the white line intensity of their catalyst, and the position of XANES derivative maximum are consistent with those for Cu²⁺ species. The obtained values of Cu-O bond lengths (Table S5) are also consistent with those for Cu²⁺ species.

Authors also do not provide any evidence that the catalyst in the presence of phosphate is indeed less prone to the reduction, since operando XAS data are provided only for the catalyst in the presence of phosphate. XAS data for the control sample reduced without phosphate needs to be provided to determine whether this statement is really valid.

Authors also state that “the high-valence Cu species underwent a rapid reduction at the first 15 mins corresponding to precatalyst dissolution (Fig. 2f)”. However, I do not see any evidence in the presented data that the catalyst is dissolved and re-deposited. From provided XAS data, as authors state themselves, it seems that the catalyst is directly transformed from the initial state into the final metallic state, without any intermediates. Do authors see any change in Cu XAS signal intensity (before normalization) that could confirm that the catalyst is indeed dissolved?

Further authors claim that the formation rate of Cu-Cu bonds at the end of the reduction is slower than at the beginning, while the contribution of Cu-O bond changes linearly through the entire process (Figure 3efg). The linear change in the catalyst composition is strange, since normally one would expect that the trends should be more or less exponential-like. Authors should comment on this in more detail. Regarding the statement that the contribution of Cu-O bonds changes differently than that of Cu-Cu bonds, I do not think it is true, and is just a matter of how one fits the Cu-O coordination number values, and their uncertainties. One can see from Figure 3g, that the final four points, if considered separately, would produce a different trendline slope, thus the evolution of Cu-O bonds is similar to that of Cu-Cu bonds.

Overall, quantitative XAS data analysis is not properly reported. It is not explained, what standard spectra were used for linear combination fits of XANES data. When reporting EXAFS fitting results in Suppl. Table 5, authors provide uncertainties only for coordination numbers, but not for other fitting variables (interatomic distances, delta E factors, sigma² factors). Authors also use different delta E factor for different paths, which is a bad practice, and is also unnecessary here, since the obtained delta E values for Cu-O and Cu-Cu values are similar anyway. Authors should also report fit R-factors or some other fit quality metrics.

Considering that their precatalyst has a structure similar to that of $\text{Cu}_2(\text{OH})_3\text{Cl}$, have authors tried to include in the fit Cu-Cl contributions? These contributions could affect significantly the results of fitting for Cu-Cu bonds.

In the main text, authors state that in the EXAFS results “the Cu-O bonds at $\sim 1.6 \text{ \AA}$ were slowly transitioned

to metallic Cu-Cu bonds at $\sim 2.3 \text{ \AA}$.” I guess they mean here the positions of corresponding peaks in Fourier-transformed EXAFS, not the corresponding bond lengths. This should be clarified.

The presented wavelet transforms in Figure 2 in the main text do not provide any additional information, and can be moved to Supplementary Information, in my opinion.

To conclude, I believe the paper cannot be published in its current form. Major revision is necessary.

Manuscript ID: NCOMMS-23-35592

“In situ copper faceting enables efficient CO₂/CO electrolysis”

Italic Black Text = Reviewer Comments

Black Text = Author Response

Blue Text = Modifications to the Manuscript

Reviewer 1

General comments:

In this study, the authors investigate by DFT calculations the effect of adsorbates (CO & OH⁻) on the surface energies of Cu(111) and Cu(100). According to the calculation results, the author proposed OH⁻ adsorption may not only decrease the surface energy of the system but also preferably promote the growth of Cu(100). Meanwhile, they also use Wulff construction calculation to estimate the ratio of Cu(100) and Cu(111) of a copper crystal with different coverages of CO and OH⁻. The results of these DFT analyses contribute to our understanding of the CO₂ reduction reaction on high-valence copper-derived catalyst systems. This study is probably publishable after the authors address the questions listed below:

R1Q1

1. Since the author assumed that there is no adsorbate-adsorbate interaction when calculating the surface energy, but the author did not provide the information (slab size, atomic layers) of the slab models in the computational details; that is, can the slab size of Cu(100) and Cu(111) used by the author in the calculation support the above hypothesis? Therefore, I suggest the author provide enough model details and structure diagrams for confirmation.

Response: We thank the reviewer for this input. We have now provided these details (see **new Supplementary Figs. 1-6**). Specifically, the slab sizes of Cu(100) and Cu(111) were 7.67 Å × 7.67 Å ($\alpha = 90^\circ$) and 7.67 Å × 7.67 Å ($\alpha = 120^\circ$), respectively, in which four atomic layers were applied (**new Supplementary Fig. 1**).

new Supplementary Fig. 1 | Model structure of (a) Cu(100) and (b) Cu(111).

To assess the adsorbate-adsorbate interaction, we calculated the average adsorption energy (E^{avg}) at a given surface coverage (θ)¹, as shown in **new Supplementary Fig. 2**.

For *CO adsorption, the change of E^{avg} is negligible (< 0.1 eV). E^{avg} changes linearly with the increase of OH^- coverage but the slopes of two fitted lines are nearly identical, indicating that the adsorbate-adsorbate interaction does not affect the relative trend of surface energies between Cu(100) and Cu(111).

new Supplementary Fig. 2 | E^{avg} of (a) *CO and (b) OH^- on Cu(111) and Cu(100) at various coverage. We exclude the adsorbate-adsorbate interaction as it does not affect the relative trend of surface energies between Cu(100) and Cu(111).

new Supplementary Fig. 3 | The top and side views of adsorption structure of Cu(100) with different *CO coverage.

new Supplementary Fig. 4 | The top and side views of adsorption structure of Cu(100) with different OH⁻ coverage.

new Supplementary Fig. 5 | The top and side views of adsorption structure of Cu(111) with different *CO coverage.

new Supplementary Fig. 6 | The top and side views of adsorption structure of Cu(111) with different OH⁻ coverage.

We have added the related details to the revised manuscript in *Paragraph 1 of Page 5*. “We began by assessing the surface coverage effect of CO* and OH⁻ on the surface energies of Cu(111) and Cu(100) using DFT (Fig. 1c, Supplementary Figs. 1-6), in which there is no adsorbate-adsorbate interaction when calculating the surface energy.”

R1Q2

2. The authors should provide the adsorption structure diagrams corresponding to Table S1 and Table S2.

Response: We have now provided these details in the revised SI (see **new Supplementary Figs. 3-6**).

R1Q3

3. Although Tables S1 and S2 give the surface energy data for different adsorbates (CO and OH⁻) coverages on Cu(100) and Cu(111), if the authors can provide some electronic analysis to discuss the trends of surface energies on these surfaces would be beneficial.

Response: We have now calculated the projected density of state (PDOS) about Cu(111) and Cu(100) with different *CO and OH⁻ coverage. We found that the *d* band center of copper downshifts with the increased adsorbate coverage (see **new Supplementary Fig. 8**), indicating that the *CO and OH⁻ species could stabilize the surface.

new Supplementary Fig. 8 | The projected density of state (PDOS) of top Cu atom on Cu(111) with different (a) *CO coverage and (b) OH⁻ coverage; PDOS of top Cu atom on Cu(100) with different (c) *CO coverage and (d) OH⁻ coverage.

We have added the below description to the revised manuscript in *Paragraph 2 of Page 5*.

“The projected density of states about Cu(111) and Cu(100) with different *CO and OH⁻ coverage were calculated to provide the electronic analysis for the adsorbates’ effect on surface energies. We found that the *d* band center of copper downshifts with increased adsorbate coverage (Supplementary Fig. 8), indicating that the *CO and OH⁻ species could stabilize the surface.”

R1Q4

4. Since the authors have added additional valence electrons to the simulation model in the DFT calculations, does this affect the dipole moment and magnetic property? If so, did the authors consider corresponding dipole moment corrections and magnetism in calculations?

Response: Using DFT we found that additional electrons of OH⁻ do not affect the dipole moment nor the magnetic property. Specifically, we calculated the dipole moment and work function under 1/9 ML and 2/9 ML of OH⁻ coverage on Cu(100), in which values of dipole moment and work function did not change at different OH⁻ coverage (**new Supplementary Tables 2 and 3, and new Supplementary Fig. 7**). We also performed the spin polarization calculation (related calculation tag is ISPIN=2) using 1/9 ML of OH⁻ coverage as an example and found a similar energy of the DFT model with or without spin polarization (**new Supplementary Table 4**).

new Supplementary Table 2 | The dipole moment of Cu(100) and Cu(111) with different OH⁻ coverage.

Dipole moment (electrons × Angstrom)	1/9 ML	2/9 ML
Cu(100)	-0.491883	-0.491883

new Supplementary Table 3 | Calculation of work function of Cu(100) with 1/9 ML and 2/9 ML of OH⁻ coverage.

OH ⁻ coverage	1/9 ML	2/9 ML
E-fermi (eV)	1.8345	1.8345
Electrostatic potential energy (eV)	4.94017	4.94017
Work Function (eV)	3.10567	3.10567

new Supplementary Fig. 7 | The work function of Cu(100) with 1/9 ML and 2/9 ML of OH⁻ coverage.

new Supplementary Table 4 | The energy of 1/9 ML of OH⁻ coverage on Cu(100) with/without spin polarization.

	System Energy (eV)
with spin polarization	-132.81613
without spin polarization	-132.81613

We have added the related details to the revised manuscript in *Paragraph 2 of Page 5*. “Additional electrons of OH⁻ do not affect the dipole moment and magnetic property of Cu (Supplementary Fig. 7 and Supplementary Tables 2-4).”

R1Q5

5. Although the authors found by the results of DFT calculations that when CO and OH⁻ coverage reaches 4/9 ML, the Cu(100) part will increase by 95% relative to the copper without intermediates, can the experimental conditions correspond to the results of the DFT simulation? If not, is such a coverage assumption reasonable? The authors should correlate the DFT simulation conditions with the experimental results.

Response: We have conducted additional XRD and electrochemical OH⁻ adsorption measurements of different Cu catalysts derived from various phosphate-doped precatalysts (**new Supplementary Figs. 15b and 22b-c, and new Supplementary Table 8**). Results show that the addition of phosphate in the precatalyst facilitates the Cu(100) growth and the portion of Cu(100) of Cu(100)-rich catalyst derived from CuP_{0.4} or CuP_{0.6} is increased by ~95% compared to that of Cu(111)-rich catalyst derived from precatalysts without phosphate, consistent with our DFT simulations.

new Supplementary Fig. 15b | XRD patterns of different Cu catalysts derived from various phosphate-loaded precatalysts. # represents the diffraction peak of carbon paper substrate.

new Supplementary Table 8 | The Cu(100)/(Cu(100)+Cu(111)) ratio of different Cu catalysts derived from various phosphate-loaded precatalysts calculated from the XRD patterns in Supplementary Fig. 15b.

P/Cu ratio in precatalyst	Area of Cu(111)	Area of Cu(100)	Cu(100)/(Cu(100)+Cu(111))
0.6	0.1447	0.1969	0.577
0.4	0.0916	0.1979	0.684
0.2	0.1999	0.2156	0.416
0	0.3221	0.1365	0.298

new Supplementary Fig. 22 | **a**, CV curves collected in N₂-purged 1.0 M KOH for Cu(100)-rich and Cu(111)-rich catalysts. **b**, CV curves collected in N₂-purged 1.0 M KOH for different Cu catalysts derived from various phosphate-loaded precatalysts. **c**, The corresponding Cu(100)/(Cu(100)+Cu(111)) ratio obtained from the CV curves.

We have expanded the related discussion in the revised manuscript in *Paragraph 1 of Page 8*.

“These results suggest that the addition of phosphate in the Cu precatalyst favors the growth of Cu(100) during CO₂R and the portion of Cu(100) of derived Cu catalyst is increased by ~95% compared to that of a previous Cu catalyst synthesized in situ¹³ (Supplementary Table 7) and that of the bare Cu control catalyst (Supplementary Figs. 15b and 22b-c, and Supplementary Table 8), consistent with our DFT simulations.”

Reviewer 2

Comments:

This paper reported the copper faceting strategy for CO₂/CO electrolysis. The fundamental study and experiments are well performed. However, the novelty of this MS does not seem sufficient for prestigious journal like NC, considering numerous publications on tuning Copper facets, including a few from the same group, such as Nat Catal 3, 98-106 (2020). In addition, the experiments and procedures developed in this study do not add any new knowledge to the existing in the field.

The authors claim high stability but Cu has been reported to go under morphology changes under cathodic conditions in the literature (e.g., Angew.Chem.Int. Ed. 2018, 57,6192-6197), therefore the authors need to provide reasons for their stability. the highly stable cell voltage at a current density as high as 300 or 500 mA/cm² needs to be elaborated since the data reported in the literature usually comes with a level of fluctuating (which can be reasonable if current is quite high).

Response: We thank the reviewer for the detailed input. We now compare this work with the previous study² as suggested (see **new Supplementary Table 7**).

new Supplementary Table 7 | Comparison of this work with a previous work¹⁰ that reported the in situ growth of Cu(100) using *CO.

	This work	Ref. ¹⁰	
Catalyst synthesis	In situ reduction of phosphate-doped precatalyst	Electrodeposition of Cu from Cu(II) ditartrate	
Cu(100) promoter	*CO&OH ⁻	*CO	
Cu(100)/(Cu(111)+Cu(100))	0.684	0.287 [†]	
MEA-CO₂R performance	FE_{C2+}	83%	60%
	j_{C2+}	415 mA cm ⁻²	180 mA cm ⁻²
	EE_{C2+}	25.5%	18.6%
MEA-COR performance	FE_{C2+}	93%	not available
	j_{C2+}	465 mA cm ⁻²	not available
	EE_{C2+}	37%	not available
	SPCE_{C2+}	95%	not available

[†]This value is calculated based on the XRD pattern of Cu-CO₂ catalyst (63s) reported in Supplementary Fig. 18a of Ref. 10.

Briefly, an increase of *CO coverage decreases the surface energies of Cu(100) and Cu(111), and the growth of Cu(111) surpasses Cu(100) formation with *CO coverage ranging from 0 ML to 4/9 ML (**Fig. 1c**). This result is in line with our previous work² showing that Cu catalyst produced from in situ reduction of Cu(II) ditartrate results in limited Cu(100) exposure, in which the Cu(100)/(Cu(100)+ Cu(111)) ratio is only 0.287.

In this work, we report a new phosphate-doping strategy to enable co-adsorption of CO&OH⁻ on Cu, promoting the growth of Cu(100) and leading to a high Cu(100)/(Cu(100)+Cu(111)) ratio of up to 0.684 (**new Supplementary Fig. 15b and new Supplementary Table 8**). This represents a ~100% increase of Cu(100) portion in our Cu(100)-rich catalyst compared to those in bare Cu and the Cu-CO₂ catalyst reported in literature².

new Supplementary Fig. 15b | XRD patterns of different Cu catalysts derived from various phosphate-loaded precatalysts. # represents the diffraction peak of carbon paper substrate.

new Supplementary Table 8 | The Cu(100)/(Cu(100)+Cu(111)) ratio of different Cu catalysts derived from various phosphate-loaded precatalysts calculated from the XRD patterns in Supplementary Fig. 15b.

P/Cu ratio in precatalyst	Area of Cu(111)	Area of Cu(100)	Cu(100)/(Cu(100)+Cu(111))
0.6	0.1447	0.1969	0.577
0.4	0.0916	0.1979	0.684
0.2	0.1999	0.2156	0.416
0	0.3221	0.1365	0.298

We echo the reviewer that Cu nanocubes comprising 100% Cu(100) is not stable during CO₂R as reported in *Angew. Chem. Int. Ed.* 2018, 57, 6192-6197 (now cited as Ref. 11 in the revised Manuscript), which often restructures to a composite of Cu(100) and Cu(111), resulting in a low portion of Cu(100) and thus limited CO₂R performance. Instead, we seek to promote the growth of stable Cu(100) from in situ CO₂ reduction, as demonstrated in our previous work². In this work, we steer the growth of Cu toward stable Cu(100) formation benefitting from the co-adsorption of CO and OH⁻ derived from electrolysis of CO₂ and H₂O. The resulting Cu catalyst with a high portion of Cu(100) helps achieve >80% FE at 500 mA cm⁻² toward C₂₊ production from both CO₂ and CO electrolysis, outperforming literature benchmarks reported in membrane-

electrode assembly devices. This work bridges the gap between nanocube and in situ copper faceting for efficient CO₂/CO electrolysis.

In addition, we have revised **Fig. 4b and 4e** by adjusting the range of full-cell potential. We can see clearly that the cell voltages at high current density present a level of fluctuation.

revised Fig. 4 | Electrochemical CO₂/CO reduction performance of the Cu(100)-rich catalyst in a MEA electrolyzer. a, FEs for all products at various current densities and related cell potentials from CO₂ reduction. **b**, Extended CO₂ electrolysis using the Cu(100)-rich catalyst at 300 mA cm⁻². Operating conditions: 0.1 M KHCO₃ anolyte at a flow rate of 20 mL min⁻¹ and an average CO₂ inlet flow rate of ~75 sccm cm⁻² at atmospheric temperature and pressure conditions. **c**, FEs for all products at various current densities and related cell potentials from CO reduction. **d**, FEs as well as SPCEs

toward C₂₊ products on the Cu(100)-rich catalyst at 500 mA cm⁻² with different CO flow rates. **e**, Extended CO electrolysis using the Cu(100)-rich Cu catalyst at 500 mA cm⁻². The Cu(100)-rich catalyst delivers an average FE_{C₂₊} of 86%, C₂₊ current densities of 432 mA cm⁻², EE_{C₂₊} of 37%, and SPCE_{C₂₊} of 94% throughout. Operating conditions: 1 M KOH analyte at a flow rate of 20 mL min⁻¹ and an average CO inlet flow rate of ~2 sccm cm⁻² at atmospheric temperature and pressure conditions.

The suggested references are now cited as *Ref. 13* and *11*. We have also included the related discussion in the revised manuscript as follows:

(Paragraph 1 of Page 3 in the revised Manuscript)

“To date, previous works in facet-controlled synthesis of Cu(100) catalysts mostly relied on colloidal synthesis using capping agents that modulate the relative energy of facets during synthesis. However, organic additives often play impact the catalytic performance and the resulting well-defined Cu catalysts such as Cu(100) nanocubes are prone to reconstruct during electrolysis.”

(Paragraph 1 of Page 8 in the revised Manuscript)

“These results suggest that the addition of phosphate in the Cu precatalyst favors the growth of Cu(100) during CO₂R and the portion of Cu(100) of derived Cu catalyst is increased by ~95% compared to that of a previous Cu catalyst synthesized in situ (Supplementary Table 7) and that of the bare Cu control catalyst (Supplementary Figs. 15b and 22b-c, and Supplementary Table 8), consistent with our DFT simulations.”

(Paragraph 3 of Page 10 in the revised Manuscript)

“This performance (EE × current) is double that of reported results in neutral MEA-CO₂R systems (Supplementary Fig. 28 and Supplementary Tables 7 and 11), indicating the role of Cu(100)-rich sites in accelerating C-C coupling.”

Reviewer 3

General comments:

The manuscript “In situ copper faceting enables efficient CO₂/CO electrolysis” by Yao et al reports on the enhanced CO₂RR to C₂₊ products, observed on copper catalysts in the presence of phosphate ligands. Authors attribute it to the enhanced formation of Cu(100) facets during the CO₂RR in the presence of OH adsorbates, which itself seems to be facilitated by the presence of phosphate ligands. Authors also claim that their pre-catalyst features an unusually high valence state for Cu, and the presence of phosphate ligand helps to retain the high valence state of copper during the electrolysis.

To conclude, I believe the paper cannot be published in its current form. Major revision is necessary.

R3Q1

1. While these claims are intriguing, unfortunately, I do not see strong evidences for most of them. First, the central claim of the manuscript is that the presence of phosphate stimulates formation of Cu(100) facets over Cu(111). However, nowhere in the manuscript I can see a quantitative analysis that this is indeed the case. Authors only claim that “XRD and dark-field TEM results showed the predominant Cu(100) over Cu(111) facets in the resulting HVCD Cu catalyst (Figs. 2d and 2h, and Supplementary Fig. 11). By contrast, a Cu(111)-rich catalyst was obtained by reducing the phosphate-free Cu precatalyst under the same conditions (Fig. 2d and Supplementary Figs. 12-13).” From the provided images I could agree that the morphology of the catalyst after the experiment are different, if the CO₂RR is carried with or without phosphate. However, I do not see any convincing, quantitative analysis that would evidence that indeed one of these catalysts have more Cu(100) facets than the other. Similarly, authors also state that “Additional electrochemical OH⁻ adsorption measurements further indicate that Cu(100)-rich and Cu(111)-rich catalysts were derived from precatalysts with and without phosphate ligands, respectively (Supplementary Fig. 14).” However, it is not clear to me how such quantitative statement can be easily extracted from Figure S14.

Response: We thank the reviewer for the comments. To enable a quantitative analysis of Cu(100)/Cu(111) ratio, we performed additional XRD and electrochemical OH⁻ adsorption measurements of Cu catalysts derived from various phosphate-loaded precatalysts (**new Supplementary Figs. 15b and 22b-c, and new Supplementary Table 8**). The results show that the addition of phosphate in the precatalyst facilitates the Cu(100) growth and the portion of Cu(100) of Cu(100)-rich catalyst derived from CuP_{0.4} or CuP_{0.6} is increased by ~95% compared to that of Cu(111)-rich catalyst derived from precatalysts without phosphate.

new Supplementary Fig. 15b | XRD patterns of different Cu catalysts derived from various phosphate-loaded precatalysts. # represents the diffraction peak of carbon paper substrate.

new Supplementary Table 8 | The Cu(100)/(Cu(100)+Cu(111)) ratio of different Cu catalysts derived from various phosphate-loaded precatalysts calculated from the XRD patterns in Supplementary Fig. 15b.

P/Cu ratio in precatalyst	Area of Cu(111)	Area of Cu(100)	Cu(100)/(Cu(100)+Cu(111))
0.6	0.1447	0.1969	0.577
0.4	0.0916	0.1979	0.684
0.2	0.1999	0.2156	0.416
0	0.3221	0.1365	0.298

new Supplementary Fig. 22 | **a**, CV curves collected in N₂-purged 1.0 M KOH for Cu(100)-rich and Cu(111)-rich catalysts. **b**, CV curves collected in N₂-purged 1.0 M KOH for different Cu catalysts derived from various phosphate-loaded precatalysts. **c**, The corresponding Cu(100)/(Cu(100)+Cu(111)) ratio obtained from the CV curves.

We have expanded the related discussion in the revised manuscript in *Paragraph 1 of Page 8*.

“These results suggest that the addition of phosphate in the Cu precatalyst favors the growth of Cu(100) during CO₂R and the portion of Cu(100) of derived Cu catalyst is increased by ~95% compared to that of a previous Cu catalyst synthesized in situ (Supplementary Table 7) and that of the bare Cu control catalyst (Supplementary Figs. 15b and 22b-c, and Supplementary Table 8), consistent with our DFT simulations.”

R3Q2

2. *The claim that the pre-catalyst features Cu sites in oxidation state higher than 2+ is also not supported by the data, in my opinion. Note that this authors' conclusion is based on XAS data, where authors compare the Cu K-edge XAS spectra for their catalyst with those of metallic Cu, Cu₂O and CuO. However, the shape of XAS features depends not only on the Cu oxidation state, but also on the local structure. If authors would compare the XAS data for their catalyst with, e.g., the reference spectrum for Cu(OH)₂, where Cu is obviously in Cu²⁺ state, they would see that the white line intensity of their catalyst, and the position of XANES derivative maximum are consistent with those for Cu²⁺ species. The obtained values of Cu-O bond lengths (Table S5) are also consistent with those for Cu²⁺ species.*

Response: We echo with the reviewer that Cu species in the precatalyst have 2+ oxidation state (see **new Fig. 2a and 2b**), in which the blueshift of XANES derivative maximum in the precatalyst (see **new Supplementary Fig. 13**) is due to the ligand effect from phosphate doping.

new Fig. 2a-b | a,b, The Cu K-edge XANES (a) and Fourier-transformed EXAFS (b) spectra of Cu precatalyst and standards (Cu foil, Cu₂O, CuO, and Cu(OH)₂).

Supplementary Fig. 13 | **a**, The Cu K-edge XANES derivative spectra of Cu precatalyst and standards (Cu foil, Cu₂O, CuO, and Cu(OH)₂), in which the blueshift of XANES derivative maximum in the precatalyst is due to the ligand effect from phosphate doping. **b,c**, Wavelet transform of the Cu K-edge EXAFS of **(b)** precatalyst and **(c)** CuO standard.

We have revised the related discussion in the revised manuscript in *Paragraph 2 of Page 7*.

“The X-ray absorption near edge structure (XANES) and its first derivative, Fourier-transformed extended X-ray absorption fine structure (EXAFS), together with the wavelet transform contour plot spectra suggested that the precatalyst exhibits a Cu structure akin to CuO (Figs. 2a and 2b, Supplementary Fig. 13).”

R3Q3

3. *Authors also do not provide any evidence that the catalyst in the presence of phosphate is indeed less prone to the reduction, since operando XAS data are provided only for the catalyst in the presence of phosphate. XAS data for the control sample reduced without phosphate needs to be provided to determine whether this statement is really valid.*

Response: We have performed in situ Cu K-edge XAS of the control sample without phosphate (see **new Supplementary Fig. 27**), in which the portion of metallic Cu in the control sample exceeds 60% within 6 mins of electrolysis. Hence, the reduction of control sample without phosphate is much faster than the reduction of precatalyst with phosphate (see **Fig. 3**).

new Supplementary Fig. 27 | **a, b**, Time-dependent Cu K-edge XANES (**a**) and Fourier-transformed EXAFS (**b**) of control Cu precatalyst without phosphate, the test was performed in CO₂-flowed 0.1 M KHCO₃ electrolyte at -1.1 V vs RHE over the course of 30-min reduction time; ocp stands for open-circuit potential. **c**, The corresponding XANES fitting spectra in (**a**), solid lines represent the experiment data and circles represent the linear combination fit spectra. **d**, The percentage of metallic Cu at different reduction times. Values are extracted from linear combination fit using the XANES spectra of control Cu precatalyst at ocp and metallic Cu with the weighting factors as fit parameters.

We have expanded the related discussion in the revised manuscript in *Paragraph 1 of Page 10*.

“We also performed in situ Cu K-edge XAS analysis of the control Cu precatalyst without phosphate (Supplementary Fig. 27); it shows a much faster Cu reduction compared to the Cu precatalyst with phosphate.”

R3Q4

4. Authors also state that “the high-valence Cu species underwent a rapid reduction at the first 15 mins corresponding to precatalyst dissolution (Fig. 2f)”. However, I do not see any evidence in the presented data that the catalyst is dissolved and re-deposited. From provided XAS data, as authors state themselves, it seems that the catalyst is directly transformed from the initial state into the final metallic state, without any intermediates. Do authors see any change in Cu XAS signal intensity (before normalization) that could confirm that the catalyst is indeed dissolved?

Response: Before normalization, the edge jump of Cu K-edge XAS drastically decreases at the first 15 mins and then remains steady (see **new Supplementary Fig. 26**), indicating that the catalyst is dissolved and re-deposited.

new Supplementary Fig. 26 | Time-dependent Cu K-edge XAFS spectra (before normalization) of Cu precatalyst with phosphate addition, in which the edge jump of XAFS spectrum drastically decreases at the first 15 mins and then remains steady at the later stage, indicating that the catalyst is dissolved and re-deposited. The test was performed in CO₂-flowed 0.1 M KHCO₃ electrolyte at -1.1 V vs RHE over the course of 30-min reduction time; ocp stands for open-circuit potential.

We have added the related discussion in the revised manuscript in *Paragraph 3 of Page 9*.

“We then fit the XANES spectra using a linear combination method (Figs. 3d and 3e, Supplementary Table 9). We found that the catalyst evolution went through an

exponential-like process: the Cu species underwent a rapid reduction at the first 15 mins corresponding to precatalyst dissolution (Fig. 2e, Supplementary Fig. 26); then at the catalyst redeposition stage, the Cu reduction assisted by co-adsorption of CO&OH⁻ was relatively slow (Figs. 2f and 3a, Supplementary Fig. 26).”

R3Q5

5. Further authors claim that the formation rate of Cu-Cu bonds at the end of the reduction is slower than at the beginning, while the contribution of Cu-O bond changes linearly through the entire process (Figure 3efg). The linear change in the catalyst composition is strange, since normally one would expect that the trends should be more or less exponential-like. Authors should comment on this in more detail. Regarding the statement that the contribution of Cu-O bonds changes differently than that of Cu-Cu bonds, I do not think it is true, and is just a matter of how one fits the Cu-O coordination number values, and their uncertainties. One can see from Figure 3g, that the final four points, if considered separately, would produce a different trendline slope, thus the evolution of Cu-O bonds is similar to that of Cu-Cu bonds.

Response: Motivated by the reviewer’s comment, we have re-performed the EXAFS fitting (see **new Fig. 3e-g** and **new Supplementary Table 10**). The evolution of Cu-Cu and Cu-O indeed follows an exponential-like trend.

new Fig. 3e-g | e, The percentage of metallic Cu at different reduction times. Values are extracted from linear combination fit using the XANES spectra of Cu precatalyst at ocp and metallic Cu as standards with the weighting factors as fit parameters. **f, g**, Coordination numbers of Cu-Cu (**f**) and Cu-O (**g**) at different reduction times extracted from the Cu K-edge EXAFS fittings (see Supplementary Table 10).

new Supplementary Table 10 | EXAFS fitting results for CuP_{0.4} during CO₂R over the course of 30-min reduction time. Data range $k = 3-11 \text{ \AA}^{-1}$, amplitude reduction factor $S_0^2 = 0.8$. We referred to a previous work³² for the fitting principle. Briefly, the coordination numbers (N) were fixed to the expected values listed in cif files of Cu₂(OH)₃Cl and Cu foil, bond distances (R) and the Debye Waller factor (σ^2) for each cell were determined. Then the Debye Waller values were fixed to calculate N. Numbers marked with * are fixed according to the information in the cif file. Bolded and unbolded scatter paths are from Cu₂(OH)₃Cl and metallic Cu, respectively.

Time	Scatter path	CN	R (Å)	σ^2 (Å ²)	R _f
0 min (ocp)	Cu-O	1.54±0.63	1.96±0.03	0.02536*	0.94%
	Cu-O	2.09±0.17	2.00±0.05	0.00269*	
	Cu-Cl	1.48±0.76	2.71±0.20	0.03381*	
	Cu-Cu	2.15±0.34	3.03±0.10	0.00799*	
	Cu-Cu	2.26±0.65	3.21±0.21	0.00854*	
	Cu-Cu	2.13±0.49	3.42±0.06	0.00750*	
3 min	Cu-O	1.14±0.78	1.95±0.04	0.02536*	0.58%
	Cu-O	2.28±0.22	1.98±0.03	0.00269*	
	Cu-Cu	1.53±0.23	2.50±0.06	0.00953*	
	Cu-Cu	2.44±0.69	3.00±0.13	0.00799*	
	Cu-Cu	2.28±0.49	3.22±0.19	0.00854*	
	Cu-Cu	2.37±0.84	3.44±0.04	0.00750*	
6 min	Cu-O	1.00±0.36	1.88±0.07	0.02536*	2.77%
	Cu-O	2.06±0.82	1.96±0.03	0.00269*	
	Cu-Cu	3.02±0.26	2.55±0.01	0.00953*	
	Cu-Cu	3.47±0.36	3.06±0.08	0.00799*	
	Cu-Cu	3.64±0.80	3.28±0.13	0.00854*	
	Cu-Cu	1.93±0.98	3.49±0.01	0.00750*	
9 min	Cu-O	0.95±0.73	1.88±0.07	0.02536*	1.25%

	Cu-O	1.31±0.41	1.94±0.05	0.00269*	
	Cu-Cu	4.52±0.28	2.55±0.01	0.00953*	
	Cu-O	0.85±0.23	1.84±0.11	0.02536*	
12 min	Cu-O	1.19±0.43	1.93±0.06	0.00269*	1.24%
	Cu-Cu	5.52±0.31	2.55±0.01	0.00953*	
	Cu-O	0.78±0.25	1.85±0.10	0.02536*	
15 min	Cu-O	0.91±0.47	1.92±0.07	0.00269*	1.23%
	Cu-Cu	6.46±0.35	2.55±0.01	0.00953*	
	Cu-O	0.62±0.07	1.86±0.09	0.02536*	
18 min	Cu-O	0.65±0.46	1.92±0.07	0.00269*	1.01%
	Cu-Cu	7.17±0.34	2.54±0.01	0.00953*	
	Cu-O	0.45±0.19	1.89±0.10	0.02536*	
21 min	Cu-O	0.45±0.39	1.92±0.03	0.00269*	0.81%
	Cu-Cu	7.95±0.33	2.54±0.01	0.00953*	
	Cu-O	0.07±0.01	1.81±0.14	0.02536*	
24 min	Cu-O	0.38±0.17	1.90±0.10	0.00269*	1.99%
	Cu-Cu	8.62±0.51	2.54±0.01	0.00953*	
	Cu-O	0.20±0.04	1.88±0.10	0.00269*	
27 min	Cu-O	0.20±0.04	1.88±0.10	0.00269*	1.80%
	Cu-Cu	9.33±0.45	2.54±0.01	0.00953*	
30 min	Cu-Cu	9.85±0.41	2.54±0.01	0.00953*	1.75%

We have clarified the related discussion in the revised manuscript in *Paragraph 3 of Page 9*.

“We then fit the XANES spectra using a linear combination method (Figs. 3d and 3e, Supplementary Table 9). We found that the catalyst evolution went through an exponential-like process: the Cu species underwent a rapid reduction at the first 15 mins corresponding to precatalyst dissolution (Fig. 2e, Supplementary Fig. 26); then at the catalyst redeposition stage, the Cu reduction assisted by co-adsorption of CO&OH⁻ was relatively slow (Figs. 2f and 3a, Supplementary Fig. 26). This was further confirmed by the EXAFS fitting results, in which the formation of metallic Cu-Cu bond also followed an exponential-like trend (Fig. 3f, Supplementary Table 10). The covalent Cu-O coordination exhibited a steep reduction (Fig. 3g), suggesting a relatively steady O-

leaching from the precatalyst matrix during chronoamperometry electrolysis. We also performed in situ Cu K-edge XAS analysis of the control Cu precatalyst without phosphate (Supplementary Fig. 27); it shows a much faster Cu reduction compared to the Cu precatalyst with phosphate.”

R3Q6

6. Overall, quantitative XAS data analysis is not properly reported. It is not explained, what standard spectra were used for linear combination fits of XANES data. When reporting EXAFS fitting results in Suppl. Table 5, authors provide uncertainties only for coordination numbers, but not for other fitting variables (interatomic distances, delta E factors, sigma² factors). Authors also use different delta E factor for different paths, which is a bad practice, and is also unnecessary here, since the obtained delta E values for Cu-O and Cu-Cu values are similar anyway. Authors should also report fit R-factors or some other fit quality metrics.

Response: Taking inspiration from a previous work (now cited as Ref. 37), we used the XANES spectra of Cu precatalyst at ocp and metallic Cu foil as standards for linear combination fits. We have also re-fitted the EXAFS spectra using same delta E factor for different paths, the results are listed in **new Supplementary Table 10**.

We have added the related discussion in the revised manuscript in *Fig.3 of Page 18*.
“Values are extracted from linear combination fit using the XANES spectra of Cu precatalyst at ocp and metallic Cu as standards with the weighting factors as fit parameters.”

R3Q7

7. Considering that their precatalyst has a structure similar to that of $\text{Cu}_2(\text{OH})_3\text{Cl}$, have authors tried to include in the fit Cu-Cl contributions? These contributions could affect significantly the results of fitting for Cu-Cu bonds.

Response: We refit the EXAFS spectra with the inclusion of Cu-Cl contribution. As shown in **new Supplementary Table 10**, Cu-Cl bond exists in the precatalyst but disappears during electrolysis (we obtained negative value of Cu-Cl coordination once we included Cu-Cl in the fitting), suggesting that Cu-Cl bond easily breaks to form metallic Cu-Cu bond during electrolysis.

R3Q8

8. In the main text, authors state that in the EXAFS results “the Cu-O bonds at $\sim 1.6 \text{ \AA}$ were slowly transitioned to metallic Cu-Cu bonds at $\sim 2.3 \text{ \AA}$.” I guess their mean here the positions of corresponding peaks in Fourier-transformed EXAFS, not the corresponding bond lengths. This should be clarified.

Response: We have revised Fig. 3c and clarified the relevant discussion in the revised Manuscript (Paragraph 2 of Page 9):

“This finding is supported by the EXAFS results (Fig. 3c), in which the intensity of Cu-O bond slowly decreased with the emergence of metallic Cu-Cu bond.”

revised Fig. 3c | Fourier-transformed EXAFS analysis of Cu pre-catalyst evolution with phosphate addition.

R3Q9

9. The presented wavelet transforms in Figure 2 in the main text do not provide any additional information, and can be moved to Supplementary Information, in my opinion.

Response: We have removed this figure panel from the MS and include it as **Supplementary Fig. 13**.

Response reference:

- 1 Grabow, L. C., Hvolbæk, B. & Nørskov, J. K. Understanding trends in catalytic activity: the effect of adsorbate–adsorbate interactions for CO oxidation over transition metals. *Top. Catal.* **53**, 298-310 (2010).
- 2 Wang, Y. *et al.* Catalyst synthesis under CO₂ electroreduction favours faceting and promotes renewable fuels electrosynthesis. *Nat. Catal.* **3**, 98-106 (2019).

REVIEWERS' COMMENTS

Reviewer #1 (Remarks to the Author):

The authors have revised the manuscript well according to the comments and provided sufficient experimental and/or computational data and details. The conclusion of this paper is adequately supported by the data and supplies significant new insight. Now, It can be recommended for publication.

Reviewer #3 (Remarks to the Author):

During the revision, authors addressed most of the points that I have raised. My only remaining comment is that authors' statement that the catalyst without phosphate is reduced "much faster" than the catalyst with phosphate needs to be softened. From Figure 3 and Figure S27 it seems that the catalyst without phosphate indeed is reduced somewhat faster than the catalyst with phosphate, but the difference is not that big, and both catalysts are mostly reduced within 30 min. To facilitate this comparison, the data currently shown in Figure S27d need to be shown in the same figure as the data currently shown in Figure 3e.

Manuscript ID: NCOMMS-23-35592A

“In situ copper faceting enables efficient CO₂/CO electrolysis”

Italic Black Text = Reviewer Comments

Black Text = Author Response

Blue Text = Modifications to the Manuscript

Reviewer 1

Comments

The authors have revised the manuscript well according to the comments and provided sufficient experimental and/or computational data and details. The conclusion of this paper is adequately supported by the data and supplies significant new insight. Now, It can be recommended for publication.

Response: We thank the reviewer for the assessment and support of our work for publication.

Reviewer 3

Comments

During the revision, authors addressed most of the points that I have raised. My only remaining comment is that authors' statement that the catalyst without phosphate is reduced “much faster” than the catalyst with phosphate needs to be softened. From Figure 3 and Figure S27 it seems that the catalyst without phosphate indeed is reduced somewhat faster than the catalyst with phosphate, but the difference is not that big, and both catalysts are mostly reduced within 30 min. To facilitate this comparison, the data currently shown in Figure S27d need to be shown in the same figure as the data currently shown in Figure 3e.

Response: We thank the reviewer for this input. We have softened the related statement in the revised manuscript in *Paragraph 1 of Page 10*.

“We also performed in situ Cu K-edge XAS analysis of the control Cu precatalyst without phosphate (Supplementary Fig. 27); it shows a rapid Cu reduction compared to the Cu precatalyst with phosphate.”

We have also updated Figure S27d to enable a clear comparison of the reduction process for Cu precatalyst with phosphate and control Cu precatalyst without phosphate (see new Supplementary Fig. 27d).

new Supplementary Fig. 27 | d, The percentage of metallic Cu at different reduction times for Cu precatalyst with phosphate and control Cu precatalyst without phosphate. Values are extracted from linear combination fit using the XANES spectra of Cu precatalyst at ocp and metallic Cu with the weighting factors as fit parameters.